# Role of Carrot (*Daucus carota* L.) Storage Roots in Drought Stress Adaptation: Hormonal Regulation and Metabolite Accumulation

**DOI:** 10.3390/metabo15010056

**Published:** 2025-01-16

**Authors:** Kyoung Rok Geem, Ye-Jin Lee, Jeongmin Lee, Dain Hong, Ga-Eun Kim, Jwakyung Sung

**Affiliations:** 1Department of Crop Science, Chungbuk National University, Cheong-ju 28644, Republic of Korea; roki2012@naver.com (K.R.G.);; 2Division of Soil and Fertilizer, National Institute of Agricultural Sciences, RDA, Wanju 55365, Republic of Korea

**Keywords:** ABA, auxin, carbohydrates, carrot (*Daucus carota* L.), GABA, osmolytes, proline

## Abstract

**Background:** Drought stress has become one of the biggest concerns in threating the growth and yield of carrots (*Daucus carota* L.). Recent studies have shed light on the physiological and molecular metabolisms in response to drought in the carrot plant; however, tissue-specific responses and regulations are still not fully understood. **Methods:** To answer this curiosity, this study investigated the interplay among carrot tissues, such as leaves (L); storage roots (SRs); and lateral roots (LRs) under drought conditions. This study revealed that the SRs played a crucial role in an early perception by upregulating key genes, including *DcNCED3* (ABA biosynthesis) and *DcYUCCA6* (auxin biosynthesis). The abundance of osmolytes (proline; GABA) and carbohydrates (sucrose; glucose; fructose; mannitol; and inositol) was also significantly increased in each tissue. In particular, LRs accumulated high levels of these metabolites and promoted growth under drought conditions. **Conclusions:** Our findings suggest that the SR acts as a central regulator in the drought response of carrots by synthesizing ABA and auxin, which modulate the accumulation of metabolites and growth of LRs. This study provides new insights into the mechanisms of tissue-specific carrot responses to drought tolerance, emphasizing that the SR plays a key role in improving drought resistance.

## 1. Introduction

Drought stress is a significant challenge to the growth of carrots [1,2], typically growing in cooler climates; yet, global temperatures have increased the prevalence of heat and drought stress [1,3]. Previous studies described that drought is a major abiotic factor in carrot production and reduces their quality [1,2,4,5]. Under water-deficient conditions for 138 days, carrot roots showed a 76% reduction in fresh mass, 67% in dry mass, and 38% in diameter compared to water-sufficient conditions. Prolonged water stress significantly impacted yield and quality [6]. Drought stress also shows significant changes in metabolite concentrations, including proline, glycine betaine and phenols compounds [7], and carotenoid content in susceptible genotypes decreases compared to tolerant ones [8]. Carrot root quality is influenced by reducing sugar contents, which determines the carrot’s taste [1,9,10]. Drought stress reduces sugar content, and a decline in the glycolytic pathway results in growth inhibition due to decreased respiration [10]. Consistently, the yield and productivity of carrots are reduced in drought conditions [1,2,11]. The deficit and limitation of available water adversely affects plant growth; however, the degree of impact varies among plant species depending on their physiological, biochemical, and molecular responses, as well as the methods of cultivation [1,2,12].

Various studies have explored the negative consequences of drought stress on the growth and yield of many vegetable and crop species. These studies have revealed disrupted physiological and biochemical changes and declines in photosynthetic activity, water potential, transpiration rates, germination rate, cell elongation, as well as the increased production of reactive oxygen species (ROS) in various plants including carrots [1,2,13,14]. Drought stress influences physiological, architectural, and morphological changes, which increase drought stress tolerance [15,16,17]. Drought stress is sensed by the roots and then transferred to other organs [16]. Roots develop deeper and thinner structures to enhance the absorption of water and nutrients [16,18]. In the shoot, when encountering drought stress, the leaves undergo various changes to adapt to the stress [19]. The shoot exhibits changes such as increased leaf thickness, wax production, leaf rolling, and a reduction in stomatal area and metaxylem vessel size to decrease evapotranspiration [19,20,21]. Carrots have also developed various strategies to gain tolerance to drought stress, and, of those, the regulation of osmotic potential with osmolyte compounds is remarkable [1,22,23], with an abundance of metabolites including proline, GABA, and various carbohydrates such as sucrose, fructose, glucose, mannitol, xylose, and inositol [22,24,25,26]. These osmolytes play a critical role in stabilizing cellular structures by controlling cell turgor pressure, water gradient, and water uptake adjustment. Many studies have reported osmotic adjustment mechanisms that help plants avoid or resist adverse environmental challenges [1,22]. Carbohydrates, which convert into various types of the reducing sugars, also play a role in carbohydrate metabolism for the plant’s energy source for growth and development under drought stress [1,22,27]. Despite much information, further investigation is still needed to extend the knowledge regarding the fine regulation in carrot plants under drought stress.

The maintenance of carrot growth and tolerance to drought stress is influenced by various plant hormones [28,29]. Previous studies have shown that plant hormones, such as abscisic acid (ABA), auxin, ethylene (ET), jasmonic acid (JA), and cytokinin (CK), regulate growth and tolerance to drought stress [28,30,31,32]. ABA plays a crucial role in drought stress responses in plants by regulating several key physiological and molecular mechanisms [29,33,34,35]. ABA leads to stomatal closure through the activation of the PYR/PYL/RCAR receptor complex, which inhibits PP2C phosphatases and activates SnRK2 kinases [29,36,37]. This ABA signaling reduces water loss via transpiration [38]. Additionally, ABA induces the expression of drought-responsive genes and promotes the accumulation of osmolytes like proline to enhance cellular dehydration tolerance [1,39,40]. Auxin also contributes to drought tolerance by modulating root architecture [29,41]. Under drought stress, auxin tends to enhance lateral root formation and root hair development, which increases water uptake. Auxin also interacts with ABA, where ABA affects auxin transport by modulating *PIN* and *LAX* protein expression in response to stress [29,42,43,44]. Together, ABA and auxin orchestrate a balanced response that promotes both water conservation and tolerance under drought conditions.

Despite a lot of valuable reports to the drought stress-derived physiological response in carrots, there is still undiscovered information, regarding tissue-specific adaptation responses under drought stress. We hypothesize that the SR is the core organ of carrots under drought stress. It is possible that the SR modulates other tissues with the interplay between various hormones and metabolites. Therefore, leaves (L), storage roots (SRs), and lateral roots (LRs) might quickly regulate their growth and development when faced with drought stress. To gain more insight into how the SR plays a role as a main regulator of drought stress in carrots, we focused on the transcriptional changes in plant hormones (ABA and auxin) and the abundance of metabolites (osmolytes and carbohydrates). Our results support that the SR modulates the growth and development of other tissues and is considered as a primary core of drought stress.

## 2. Materials and Methods

### 2.1. Plant Material and Drought Condition

Seeds of the carrot cultivar (*Daucus carota* L. cv. Shinheukjeon5chon) were sterilized in 5 L of distilled water containing 2.5 mL of a seed sterilizer (Kimaen, Farm Hannong, Seoul, Republic of Korea) for 24 h at room temperature and then germinated in an incubator (in darkness, at 25 °C) for 5–7 days. After germination, the seedlings were carefully transplanted into hydroponic cultivation containers (35 × 25 × 10 cm; length, width, and height) containing a 0.8× Hoagland nutrient solution. The transplanted carrot plants were grown in an incubator (VS-8407-800-0, Visionbionex, Gyeonggi, Republic of Korea) for 40 days under the following conditions: 14 h of light at 25 °C, 10 h of darkness at 25 °C, and 60% relative humidity (RH). The forty-day-old carrot plants were treated with two different water potentials, 0 and −1.5 MPa (10% of PEG-6000), in 0.8× Hoagland nutrient solution for 3 and 10 days. The carrot plants were carefully harvested, and phenotypic parameters and the dry weight were measured at 3 or 10 days after the drought treatment (DAT). The collected samples were stored at −80 °C until further analysis.

### 2.2. Analysis of Metabolites

A proline analysis was performed following the previous method described [12]. Fresh samples (0.5 mg) extracted from carrot leaves, storage roots, and lateral roots were mixed with 1 mL of 3% sulfosalicylic acid and centrifuged at 14,000× *g* for 5 min at 4 °C. The supernatant was carefully collected. The collected supernatant (500 µL) was promptly mixed with a reaction solution composed of 1 mL of glacial acetic acid and 1 mL of acidic ninhydrin solution (1.25 g ninhydrin, 30 mL glacial acetic acid, and 20 mL of 6 M orthophosphoric acid), then incubated at 96 °C for 1 h, with the reaction subsequently terminated on ice. Toluene (2 mL) was added to the reaction mixture, and the supernatant was used for proline measurement using a spectrophotometer (UV-1900i, Shimadzu, Kyoto, Japan) at 520 nm. L-proline (Sigma, St. Louis, State of Missouri, USA) served as the standard (R^2^ = 0.99). The proline content was calculated using the following formula: Proline content (µg per g of fresh sample) = [(µg Proline/mL × mL Toluene)/(115.5 µg/µmol)]/[(g sample)/5].

Lyophilized samples (30 mg) were subjected to extraction using 1 mL of 70% MeOH with sonication for 1h. The resulting slurry was then centrifuged at 11,000× *g* at 4 °C for 10 min. Following filtration through 0.20 μm microporous membranes, 200 μL of the supernatant was placed into a 1.5 mL microtube and dried using a SpeedVac. The dried extracts were redissolved and derivatized by adding 50 μL of methoxyamine hydrochloride (20 mg mL^−1^) in pyridine. Additionally, 50 μL of fluoranthene (1000 μg mL^−1^ in pyridine) was incorporated as an internal standard, and the mixture was incubated for oximation at 30 °C for 90 min. The oximated samples underwent trimethylsilylation by adding 100 μL of N, O-bis (trimethylsilyl) trifluoroacetamide and heating at 60 °C for 30 min in preparation for the GC (Crystal 9000, Chromatec, Yoshkar-Ola, Russia)-MS (Crystal MSD, Chromatec, Russia) analysis. An aliquot of 1.0 μL was injected into the GC-MS with a 15:1 split ratio, utilizing a DB-5MS capillary column (60 m × 0.25 mm × 0.25 μm, Agilent J & W Scientific, Folsom, CA, USA). Helium was employed as the carrier gas, maintaining a constant flow rate of 1.7 mL/min. The temperature protocol included an initial setting of 50 °C for 2 min, increased to 180 °C at a rate of 5 °C/min and held for 8 min, followed by an increase to 210 °C at 2.5 °C/min, then raised to 325 °C at 5 °C/min and maintained for an additional 10 min. The injector, transfer line, and ion source temperatures were set at 280 °C, 280 °C, and 275 °C, respectively. Mass spectrometry was performed in full-scan mode with a mass range of 35–650 *m*/*z* under electron-impact ionization (70 eV). The solvent delay time was set to 14 min. Each metabolite’s identification was confirmed by comparing retention times and mass spectral data against the NIST/EPA/NIH Mass Spectral Library (version 2.0 d, National Institute of Standards and Technology, Gaithersburg, MA, USA). All metabolites were identified by matching mass fragments with the standard mass spectra in the commercial database NIST, requiring a similarity score of over 70%. The area calculated for each compound was normalized by dividing the peak area of the internal compound (fluoranthene) to yield a semi-quantitative composition of the components. The quantification of each compound was achieved against the internal standard by integrating the peak areas.

### 2.3. Quantitative Real-Time PCR

Total RNA was extracted from the leaves, storage roots, and lateral roots of carrots at 3 or 10 days after the drought treatment (0 and −1.5 MPa, respectively) using TRIzol reagent (Invitrogen, Carlsbad, CA, USA), following the manufacturer’s instructions. The purity and concentration of the extracted RNA were measured using a NanoDrop (Thermo Fisher Scientific, Madison, WI, USA) and verified on a 1.2% agarose gel. First-strand cDNA was synthesized from 1 μg of total RNA using the RT PreMix Kit with Oligo (dT) primers under the following conditions: 60 min at 45 °C for cDNA synthesis and 5 min at 95 °C for RTase inactivation. Quantitative real-time PCR was performed on a real-time PCR machine (CFX Opus 96, Bio-Rad, Hercules, CA, USA) with technical triplicates. The reaction mixture contained 1 μL of cDNA template, 2 μL of 10 mM forward and reverse primers, and 5 μL of SYBR Green Q Master Mix (Labopass, Cosmo Genetech, Seoul, Republic of Korea). The PCR conditions were as follows: an initial denaturation step at 95 °C for 5 min, followed by 50 cycles of denaturation at 95 °C for 10 s, annealing at each primer’s specific temperature for 30 s, and elongation at 72 °C for 20 s. A melting curve analysis was performed, ranging from 65 to 95 °C at a heating rate of 0.5 °C/s. The quantification method (2^–∆∆Ct^) was used as described by [45], and expression variations were estimated with technical triplicates for each cDNA sample. The carrot actin gene was used as a reference gene in the qRT-PCR. The primer sequences for qRT-PCR were designed using Primer3 software [46]. All the primers are listed in Appendix A.

### 2.4. Statistical Analysis

Statistical analyses were performed using R software (version 4.0.3), R Studio (version 1.3.1093), and GraphPad Prism software. Results are expressed as mean values with standard deviations. Differences between groups were assessed using a one-way ANOVA followed by Tukey’s honest significant difference (HSD) test. For qRT-PCR and fatty acid analysis data, statistical evaluations were carried out using SAS software (version 9.4, SAS Institute, Cary, NC, USA), with the significance determined by Fisher’s least significant difference (LSD) test at a 5% probability level.

## 3. Results

### 3.1. Carrot Growth and Root Development Under Different Water Potentials

Water is essential for plant growth, but its availability is often limited. The carrot, which is a root crop, is particularly sensitive to water deficits during root development [12]. In this study, we used the permanent wilting point (−1.5 MPa) and a control (0 MPa) using PEG-6000 to investigate the tissue-specific adaptations of carrot to drought stress. The growth and wilting of carrots were significantly affected by the −1.5 MPa (Figure 1A). A visible phenotypic change by water stress was marked in the shoot at 10 days after treatment (DAT) (Figure 1A–C). The shoot length and dry weight of carrots under −1.5 MPa were significantly reduced by 14% and 53%, respectively, compared to the control at 10 DAT (Figure 1B,C). Although the root length did not change significantly, the root color tended to be greener in −1.5 MPa (Figure 1A,D). Interestingly, the shoot and root showed a reduction in growth and development; however, the lateral root growth showed an increase under −1.5 MPa (Figure 1E), 3.2-fold higher compared to 0 MPa at 3 DAT (Figure 1E).

### 3.2. Physiological Indicators of Carrot Under Different Water Potentials

Osmolytes and carbohydrates were accumulated in carrots under limited water potential (−1.5 MPa). To obtain more insight into the water stress response of each tissue, we analyzed the accumulation of osmolytes, which are proline, pyroglutamic acid (5-oxoproline), and gamma-aminobutyric acid (GABA). Proline, an osmo-protectant, was significantly increased in leaves (L), storage roots (SRs), and lateral roots (LRs) by water stress (−1.5 MPa) at 3 DAT (12-, 2.1-, and 1.1-fold increased, respectively) and 10 DAT (2.5-, 5.85-, and 2.37-fold increased, respectively) (Figure 2). Proline showed a differences level dependent on the tissue, and it tended to accumulate the highest in L, followed by SRs and LRs (Figure 2). The 5-oxoproline was markedly increased in L and LRs, showing an elevation of 3.04- and 8.21-fold, respectively, at 10 DAT (Figure 3A), while it was reduced by 63% in SRs (Figure 3A). Moreover, GABA, known for enhancing stress tolerance, was remarkably increased in LRs, showing a 2.55-fold increase. However, GABA levels in the L and SRs decreased by 78% and 84%, respectively, at 10 DAT (Figure 3B). The accumulation of carbohydrates is important not only for the regulation of osmotic potential but also as an energy source through the TCA cycles [1,27,47]. We analyzed the accumulation of selected non-structural carbohydrates, including sucrose, glucose, fructose, mannitol, and myo-inositol (Figure 4). Sucrose was highly increased in the SRs and LRs, with a 2- and 10-fold increase, whereas the abundance in the L showed a reduction of 20% (Figure 4A). Glucose, fructose, and mannitol showed similar accumulation patterns in carrots (Figure 4B–D). The L exhibited an increased accumulation of glucose, fructose, and mannitol, with 17-, 4.2-, and 5.3-fold higher levels under −1.5 MPa (Figure 4B–D). Moreover, glucose, fructose, and mannitol were significantly increased in LRs, with 11.5-, 3.8-, and 2.38-fold increases under water stress (Figure 4B–D). In contrast, SRs showed the reduced accumulation of glucose, fructose, and mannitol, with reductions of 20%, 38%, and 20%, respectively (Figure 4B–D). Additionally, myo-inositol showed a 3.2-fold increase in LRs and a 30% decrease in SRs, with no significant difference observed in the L (Figure 4E). Our results showed that the majority of carbohydrates were significantly accumulated in the L and LRs, while SRs showed an increase in only sucrose (Figure 4).

### 3.3. ABA and Auxin Biosynthesis and Signaling in Carrots Under Different Water Potentials

To understand the tissue-specific responses of growth and development in carrot plants under water stress, we investigated the transcriptional variations of abscisic acid (ABA)- and auxin-related genes, which were directly involved in the biosynthesis and signaling of ABA and auxin (Figure 5 and Figure 6). At 3 DAT, *DcNCED3*, an ABA biosynthesis gene, was markedly reduced in both L and LRs, showing reductions of 40% and 49% under −1.5 MPa, respectively (Figure 5A,B), while it was significantly increased in SRs, indicating a 2.52-fold increase under −1.5 MPa (Figure 5C). By contrast, at 10 DAT, *DcNCED3* was obviously increased only in the L, to 7.9-fold higher; however, SRs and LRs were unchanged (Figure 5D–F). Moreover, *DcPYL9* (an ABA receptor) also showed a noticeable increase in the L, SRs, and LRs, indicated at 2.01-fold, 3.1-fold, and 2.06-fold higher compared to 0 MPa at 3 DAT. At 10 DAT, *DcPYL9* showed a significant increase in the SRs, with a 5.7-fold higher expression, a slight increase in the L with a 1.2-fold higher expression, and a slight decrease in the LRs, with a 10% reduction (Figure 5 E,F). Meanwhile, the transcript level of *DcSnRK2*, a key signaling kinase, remained unchanged at 3 DAT (Figure 5A–C). However, at 10 DAT, the *DcSnRK2* level decreased in the L, SRs, and LRs, showing reductions of 63%, 79%, and 26%, respectively (Figure 5D–F).

We also investigated the auxin-related genes. *DcYUCCA6*, an auxin biosynthesis gene, remained unchanged or was slightly reduced in the L and LRs (Figure 6A,B). However, it was significantly upregulated in the SRs, showing a 2.95-fold increase compared to 0 MPa at 3 DAT (Figure 6C). *DcLAX1*, an auxin transporter gene, showed a 2.28-fold increase in the SRs, while the L and LRs remained unchanged or slightly increased at 3 DAT (Figure 6A–C). *DcARF8*, an auxin response transcription factor, increased in the L and LRs, showing 2.3- and 3.81-fold increases, respectively (Figure 6A,B). However, *DcARF8* was reduced by 61% in the SRs (Figure 6C). By contrast, at 10 DAT, most auxin-related genes tended to decrease in all tissues of the carrot plant. *DcYUCCA6* was reduced in all tissues, showing reductions of 34%, 61%, and 66%, respectively (Figure 6D–F). *DcARF8* was reduced in the L and LRs, showing reductions of 54% and 72%, respectively, but did not significantly change in the SRs at 10 DAT (Figure 6D–F). Moreover, *DcLAX1* was highly elevated in the L, with a 5.65-fold increase (Figure 6D). However, the SRs and LRs showed 67% and 93% decreases in *DcLAX1*, respectively, at 10 DAT (Figure 6E,F). The ABA and auxin biosynthesis-related genes, which are *DcNCED3* and *DcYUCCA6*, were significantly increased only in the SRs at 3 DAT, as the plant perceived the water stress (Figure 5 and Figure 6). These results suggest that the SR is the main regulator of drought stress through the biosynthesis of ABA and auxin in carrot.

## 4. Discussion

This study employed −1.5 MPa, the wilting point, for 3 and 10 days to investigate the differences in tissue-specific responses. We focused on elucidating the physiological responses, metabolites accumulation, and hormonal regulation. The development and growth of carrot plants were significantly affected by −1.5 MPa (Figure 1A), indicating that limited water availability inhibits the growth of shoots and roots. To investigate the tissue-specific responses under water stress, we examined the leaves (L), storage roots (SRs), and lateral roots (LRs) of carrot plants subjected to 3 and 10 days of water stress. Under water stress, the plant height and dry weight were significantly reduced [48]. Our results showed that the shoot and root length were not significantly changed across the L, SRs, and LRs at 3 DAT (Figure 1); however, the extension, 10 DAT, of water stress led to a significant decrease in the length and dry weight of the shoots (Figure 1B,C). Although the root length showed no substantial difference from the control under water stress, interestingly, the LR growth increased from 3 DAT (Figure 1E). It is well known that drought stress induces LR formation in other species [34]. Our results also suggest that carrots promote the LR growth as a phenotypic response to early water stress.

Proline, an osmo-protectant, is accumulated in plants from water stress [1,7], and tends to be in greater abundance in the shoot part compared to the roots [1,12,49]. Our results also demonstrated a significant accumulation of proline in carrot plants experiencing water stress (Figure 2), indicating significantly greater proline levels. In addition, GABA is accumulated during plant responses to environmental stresses such as drought, high temperatures, waterlogging, salt, hypoxia, excessive reactive oxygen species (ROS) content, and heavy metals [24]. Our result showed that GABA was significantly accumulated in LRs (Figure 3B). Previous studies showed that proline and GABA play a role as energy adjusters in drought stress responses, and are related to ABA signaling [12,24,50]. The accumulation of carbohydrates in plant cells declines the osmotic potential, so a high concentration of carbohydrates important in plant cells have a crucial role in water stress [1,27]. Previous studies suggested that drought stress induced the accumulation of reducing sugars and sugars which serve as an osmo-regulator under water stress [47]. The selected carbohydrates were highly accumulated in LRs (Figure 4). Taken together, various metabolites tend to accumulate in LRs to promote growth and development, enabling adaptation to drought stress.

ABA triggers the metabolisms for drought response and ABA response via the PYL/PYR/RCAR-PP2C-SnRK2 cascade [51,52]. We investigated the transcriptional changes by selecting the key ABA-related signaling genes, including *DcNCED3* (an ABA biosynthesis gene), *DcPYL9* (an ABA receptor), and *DcSnRK2* (a key signaling kinase). In this study, *DcPYL9* shows a significant increase in the L, SRs, and LRs at 3 DAT (Figure 2). It is possible that the ABA signaling was elevated in all tissues as a mechanism for carrots to overcome initial water stress. However, prolonged drought stress causes significant damage, including wilted shoots and reduced carotenoid levels in SRs (Figure 1A). ABA signaling plays a crucial role in the dormancy of perennating root crops and the trade-off between plant growth and stress adaptation [53,54]. Furthermore, SRs are essential for sustaining growth and development in the subsequent year [53]. It is possible that prolonged drought stress reduced growth-related mechanisms while enhancing mechanisms of stress adaptation in SRs through the increasing *DcPYL9*. In addition, previous studies showed that roots play a role in regulating plant development under drought stress [17]. The root is the first organ to recognize drought stress and responds through morphological, anatomical, and genetic modulations [17,55]. ABA accumulates in the roots and is transported to other tissues under drought stress conditions. Additionally, ABA circulates between the root and shoot [56,57,58]. This transported ABA plays a role in closing stomata and facilitating root-to-shoot communication before the shoot recognizes the water deficiency [59]. Consequently, under drought stress, the root sends various signals, including ABA, Ca²⁺, and peptide signals, to the aboveground parts [17,60]. The transported ABA produced in roots induces shoot ABA signaling to reduce water loss 11 [58,59]. These signaling communications regulate the root-to-shoot ratio by promoting lateral root growth for water uptake and inhibiting shoot growth [17]. Our study also showed that, at an early stage, the root responds to drought stress by upregulating *DcNCED3* in SR for ABA synthesis at 3 DAT (Figure 5). This synthesized ABA may be transported to the carrot shoot to respond to drought stress. Prolonged drought stress subsequently induces ABA synthesis in the leaves at 10 DAT (Figure 5). Therefore, our results suggest that SRs play an important role in drought responses through ABA biosynthesis and communication with the shoot via ABA signaling in early drought stress.

Moreover, ABA signaling also affects auxin-related signaling by modulating the expression of PIN and LAX proteins in response to drought stress [29,42,43,44]. Auxin contributes to drought tolerance by modulating root architecture and enhancing LR formation [1,39,40]. We analyzed the expression levels of key auxin-related genes, such as *DcLAX1* (an auxin transporter gene), *DcYUCCA6* (an auxin biosynthesis gene), and *DcARF8* (an auxin response transcription factor). Our result showed that *DcYUCCA6* was increased only in SRs at 3 DAT (Figure 6). These results are consistent with *DcNCED3* and can give more strength to the possibility that the SR is responsible for water stress by synthesized ABA and auxin. ABA also serves as a regulator of drought stress by enhancing the biosynthesis of osmolytes, including proline, organic acids, and protective proteins [39,61]. It was demonstrated that ABA was positively correlated with an increased concentration of various metabolites [62]. Furthermore, our results showed that various metabolites were highly accumulated in LRs. The accumulated metabolites, such as proline, GABA, and other reducing sugars, mitigate ROS stress induced by salinity and drought stress [1,2,13]. In addition, these metabolites are involved in the regulation of water potential for drought stress tolerance by modulating osmotic potential and reducing ROS [1]. Previous studies showed that ABA produced in roots regulates the LR root growth and architecture and affects the leaves transpiration [29,59]. Therefore, our results demonstrated that SRs affect the accumulation of metabolites and promote LR growth (Figure 1) to enhance drought stress tolerance by synthesizing ABA and auxin. In conclusion, our findings suggest that SRs play an important role in drought tolerance by synthesizing ABA and auxin during the early drought response.

Further studies with SR-specific regulatory networks are required to extend our knowledge on the mechanisms enhancing the tolerance to drought stress. Currently, we are studying to understand how SRs sophisticatedly control LR development in terms of hormonal crosstalk.

## 5. Conclusion

Results of the current study imply how carrot plants respond to drought stress with different tissue-specific responses. The early response of drought stress was the upregulation of DcNCED3 (ABA) and DcYUCCA6 (auxin) genes in SRs. With an increase in drought conditions, metabolites such as proline, GABA, and soluble carbohydrates were subsequently accumulated in L, SRs, and LRs, and thus, lateral roots were significantly developed. Despite better understanding tissue-specific responses against drought in carrot plants, further studies should be extended to investigate regulatory networks between tissues with multi-omics technologies under diverse water-limited conditions.

## Figures and Tables

**Figure 1 metabolites-15-00056-f001:**
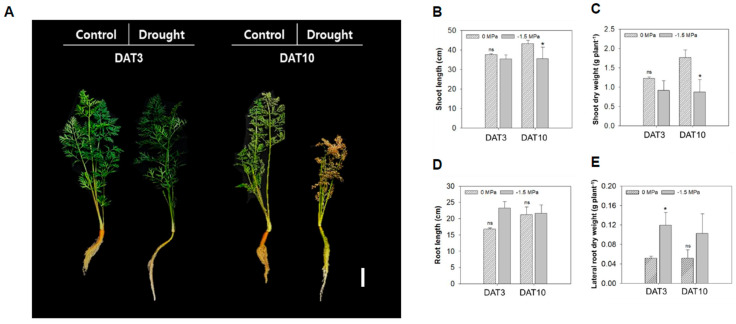
**Growth and biomass production of carrot plant under different water potentials.** (**A**) Phenotype of 40-day-old carrots (*Daucus carota L.*) treated with 0 and −1.5 Mpa (10% of PEG-6000). Scale bar = 5 cm. Measurement of (**B**) shoot length, (**C**) root length, (**D**) shoot dry weight, and (**E**) lateral root dry weight. Statistical significance was determined by a T-test: * *p* < 0.05. ns indicates non-significant difference.

**Figure 2 metabolites-15-00056-f002:**
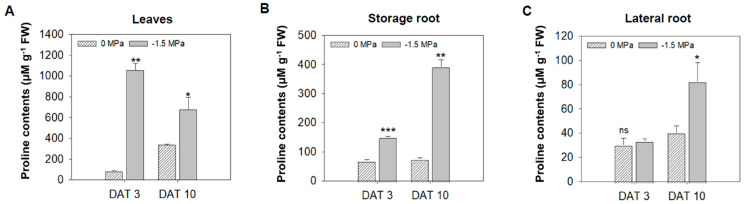
**Proline contents under different water potentials.** The content of proline was measured in the leaves (L), storage root (SR), and lateral root (LR) from 40-day-old carrots treated with 0 and −1.5 Mpa (10% of PEG-6000). Measurement of proline contents in (**A**) leaves, (**B**) storage roots, and (**C**) lateral roots at 3 and 10 DAT. Statistical significance was determined by a T-test: * *p* < 0.05; ** *p* < 0.01; *** *p* < 0.001. "ns" indicates no significant difference in proline levels in the leaf, storage roots, and lateral roots of carrots under different water potentials.

**Figure 3 metabolites-15-00056-f003:**
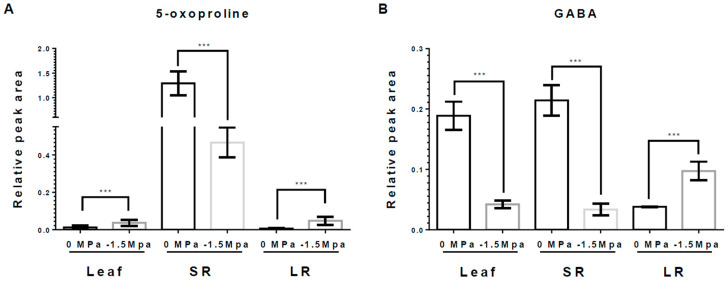
**5-Oxoproline and GABA contents in leaf, SR, and LR under different water potentials.** 5-Oxoproline and GABA were extracted from 40-day-old carrots treated with 0 and −1.5 Mpa (10% of PEG-6000) at 10 DAT. Measurement of (**A**) 5-Oxoproline contents, and (**B**) GABA contents in leaf, SR and LR at 10 DAT. Statistical significance was determined by T-test: *** *p* < 0.001 in the leaf, storage roots, and lateral roots of carrots under different water potentials.

**Figure 4 metabolites-15-00056-f004:**
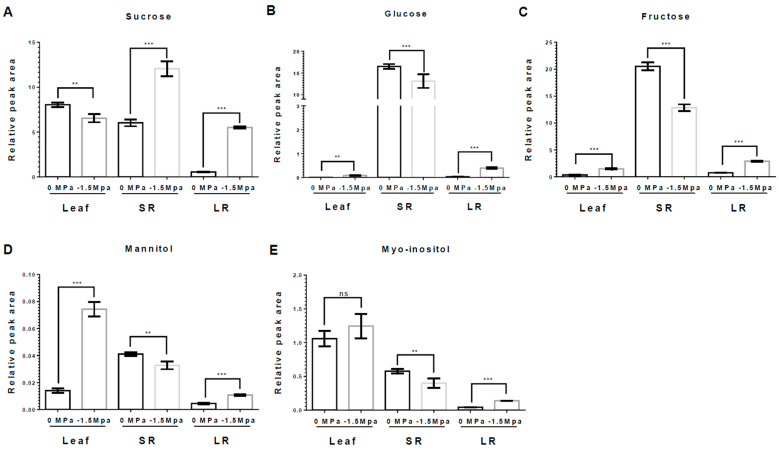
**Abundance of selected carbohydrates in leaf, SR, and LR under different water potentials.** All carbohydrates were extracted from 40-day-old carrots treated with 0 and −1.5 Mpa (10% of PEG-6000) at 10 DAT. Measurement of (**A**) Sucrose, (**B**) Glucose, (**C**) Fructose, (**D**) Mannitol, and (**E**) Myo-inositol contents in leaf, SR and LR at 10 DAT. Statistical significance was determined by T-test: ** *p* < 0.01; and *** *p* < 0.001 in the leaf, storage roots, and lateral roots of carrots under different water potentials. "ns" indicates no significant difference.

**Figure 5 metabolites-15-00056-f005:**
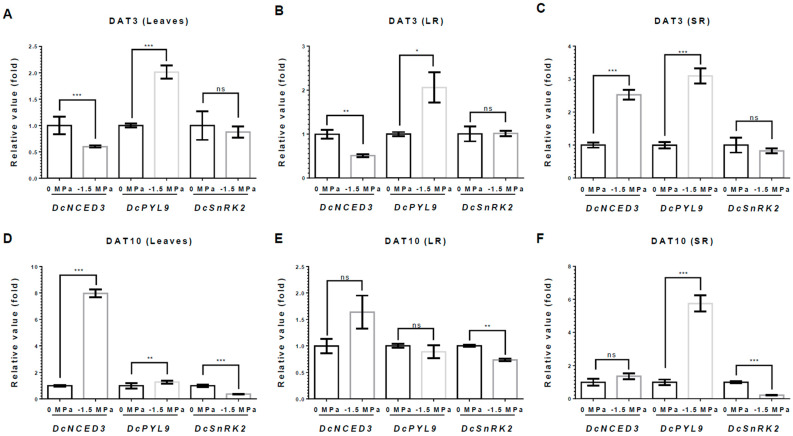
**Relative expression of ABA-related genes under different water potentials.** Total RNA was extracted from leaves(L), storage root (SR), and lateral root (LR) from 40-day-old carrots treated with 0 and −1.5 Mpa (10% of PEG-6000), and analyzed by quantitative real-time reverse transcription-polymerase chain reaction (qRT-PCR). The expression levels of *DcNCED3*, *DcPYL9*, and *DcSnRK2* were measured at 3 DAT in (**A**) L, (**B**) LR, and (**C**) SR and at 10 DAT in (**D**) L, (**E**) LR, and (**F**) SR. Statistical significance was determined by T-test: * *p* < 0.05; ** *p* < 0.01; and *** *p* < 0.001 in the leaf, storage roots, and lateral roots of carrots under different water potentials.

**Figure 6 metabolites-15-00056-f006:**
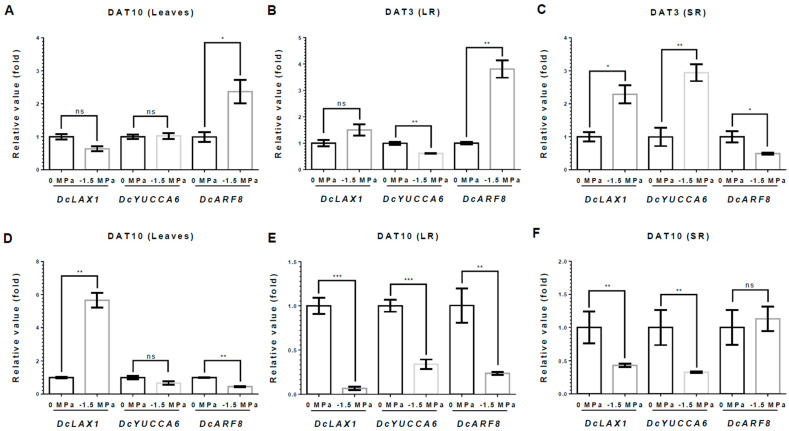
**Relative expression of Auxin-related genes under different water potentials.** Total RNA was extracted from leaves(L), storage root (SR), and lateral root (LR) from 40-day-old carrots treated with 0 and −1.5 Mpa (10% of PEG-6000), and analyzed by quantitative real-time reverse transcription-polymerase chain reaction (qRT-PCR). The expression levels of *DcLAX1*, *DcYUCCA6*, and *DcARF8* were measured at 3 DAT in (**A**) L, (**B**) LR, and (**C**) SR and at 10 DAT in (**D**) L, (**E**) LR, and (**F**) SR. Statistical significance was determined by T-test: * *p* < 0.05; ** *p* < 0.01; and *** *p* < 0.001 in the leaf, storage roots, and lateral roots of carrots under different water potentials.

## Data Availability

The raw data supporting the conclusions of this article will be made available by the authors on request.

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
