# Peer review of "Role of Carrot (Daucus carota L.) Storage Roots in Drought Stress Adaptation: Hormonal Regulation and Metabolite Accumulation"

_metabolites, 2025, doi:10.3390/metabo15010056_

Round 1
Reviewer 1 Report
Comments and Suggestions for Authors
the suggestions and recommendations are included in the word file.

the quality of the english written in the manuscript is good. however the author has written many sentences in repitition in the manuscript as highlighted in the attached word file.
Author Response
We greatly appreciate the constructive comments provided by the reviewers.
Please check the attached file for the response.
Thank you.

Reviewer 2 Report
Comments and Suggestions for Authors
This study focuses on the mechanism of tissue-specific response to drought tolerance in carrot. The work shows that SR plays a key role in enhancing drought tolerance.
Using a small and trivial set of methods, the authors showed that the amount of osmolytes and carbohydrates significantly increased in leaves (L); storage roots (SR) and lateral roots (LR); under drought condition. It was also shown that SR plays a critical role in an early perception by upregulating DcNCED3 and DcYUCCA6 .
There are several comments on the presented material:
However, from the conclusion of the work it is not entirely clear for what the obtained results are applicable and what their fundamental novelty is. It is probably worth describing this point in more detail.
There are some minor errors in the design of the manuscript.
Comments on the Quality of English Language
The English could be improved to more clearly express the research.
Author Response

(The authors gave the same response as above.)

Reviewer 3 Report
Comments and Suggestions for Authors
The article " Role of carrot (Daucus carota L.) storage roots to drought stress adaptation: hormonal regulation and metabolite accumulation" examines the effect of drought on the carrot plant. The article is interesting, but the conclusions made by the authors should be corrected.
1. The introduction should briefly list the main physiological reactions of carrots to drought. Identify the role of various organs in drought. What kind of drought is fatal for carrots?
2. It is necessary to provide information on already known works in this area concerning carrots and related species.
3. It is necessary to improve the quality of Figures 3, 4, 5 and 6.
4. How long after the onset of drought were the results shown in Figures 3 and 4 obtained? Add this information to the figure caption. Why are the results shown for only one DAT variant?
5. The statement " The accumulation of proline did not depend on the tis- 183 sues " line 183-184 does not correspond to the results shown in Figure 2. I recommend correcting this phrase.
6. " The 5-oxoproline was markedly increased in L and LR, showing elevation of 3.04-and 8.21-fold, respectively, at 10 DAT (Figure 3A), while was reduced by 63% in SR (Figure 5A)" . line 184-186. Figure 5A does not show these results. Correct.
Discussion
7. " Although previous studies have explained the responsive mechanism of carrot plant to water stress, the tissue-specific responses are still not fully understood " line 245-246. It is worth describing these mechanisms and providing relevant references.
8. It is necessary to justify the choice of using PEG to create drought. And creating -1.5 MPa.
9. The authors base their conclusions about the role of ABA and IAA during drought on the analysis of genes involved in their metabolism, transport and reception. However, changes in the levels of these hormones during drought are not shown. Is it known what happens to these hormones during drought in carrots? This information is not enough to make the results of the article convincing.
10. The authors also base their conclusions about the main role of storage roots (SR) during drought on experiments on genes associated with ABA and IAA. The results of other experiments are not so clear. I propose to reconsider the conclusion about the main role of storage roots (SR) and take into account the role of other parts of carrots.
Author Response

(The authors gave the same response as above.)

Round 2
Reviewer 2 Report
Comments and Suggestions for Authors
In order to emphasize the value of the conducted research, the authors should insert into the text of the manuscript's conclusion a formulation from their own response to the comment in the first round of review."The results of this study indicate that tissuespecific drought responses are triggered by the synthesis of ABA and auxin in storage roots. These findings are the first to explain the different tissue-specific responses and how carrot tissues respond to drought stress."
Author Response
We greatly appreciate the constructive comments provided by the reviewers.
Please check the attached file for the response. The revised manuscript is highlighted in yellow.
Thank you.

Reviewer 3 Report
Comments and Suggestions for Authors
The authors have worked to improve the manuscript. However, in my opinion, it is not sufficient.
1. The introduction needs to review the information provided about the impact of drought on carrots and other plants. In lines 28-48, the authors provided new information, but it is very general and does not make any new contribution to understanding the importance of studying drought in relation to carrots. What intensity and duration of drought is detrimental. What crop losses occur due to drought? What has already been done for carrots and published in the scientific literature. What works on related species, plants of the same Umbelliferae family are known?
2. As I already noted in the first review, the quality of figures 3, 4, 5 and 6 needs to be improved. This has not been done. The figures are small, the captions on the axes are not legible at 100% page scale.
3. I would like to draw attention to the fact that the authors did not study the change in the level of ABA and IAA in carrots during drought and do not provide information about similar experiments by other authors in the discussion. At the same time, the authors draw conclusions about the role of these phytohormones during drought in carrots. This is unacceptable and questionable.
4. I repeat my question from the first review. The authors base their conclusions about the main role of storage roots (SR) during drought on experiments on genes associated with ABA and IAA. The results of other experiments are not so clear-cut. I propose to revise the conclusion about the main role of storage roots (SR) and take into account the role of other parts of the carrot. Large metabolic changes occur in LR.
Author Response

(The authors gave the same response as above.)

Round 3
Reviewer 3 Report
Comments and Suggestions for Authors
The authors provided new information in the introduction, using literature on the effect of drought on carrots. The figures were corrected.
However, in the conclusion, the authors allow the following conclusions:
"The SR is crucial for synthesizing key hormones such as abscisic acid (ABA) and auxin, which regulate drought stress response and LR formation. During early drought conditions, SR acts as a primary source of ABA, which triggers adaptive processes in other tissues, including the leaves and LR". Lines 365-368
These conclusions are not based on the results of the authors' experiments. I believe that the article cannot be published in this form.
Author Response

(The authors gave the same response as above.)

Round 4
Reviewer 3 Report
Comments and Suggestions for Authors
Dear authors!
In the conclusion, you make unfounded conclusions.
In the work, you did not show that auxin and ABK are formed in the root, and did not trace their path to other organs of the cell, did not identify the launch of signaling systems in them. Therefore, you cannot talk about their signaling action. You are misleading readers.
The article can be published only in the case of a complete change in the conclusions, which simply speak of the facts of the identified changes.
Author Response

(The authors gave the same response as above.)
